# Pleomorphic Adenoma of the Salivary Glands and Epithelial–Mesenchymal Transition

**DOI:** 10.3390/jcm11144210

**Published:** 2022-07-20

**Authors:** Yuka Matsumiya-Matsumoto, Yoshihiro Morita, Narikazu Uzawa

**Affiliations:** Department of Oral and Maxillofacial Surgery II, Graduate School of Dentistry, Osaka University, 1-8 Yamadaoka, Suita-shi 565-0871, Osaka, Japan; matsumiya.yuka.dent@osaka-u.ac.jp (Y.M.-M.); uzawa.narikazu.dent@osaka-u.ac.jp (N.U.)

**Keywords:** pleomorphic adenoma, epithelial–mesenchymal transition, carcinoma ex-pleomorphic adenoma

## Abstract

Pleomorphic adenoma (PA) is a localized tumor that presents pleomorphic or mixed characteristics of epithelial origin and is interwoven with mucoid tissue, myxoid tissue, and chondroid masses. The literature reported that PA most often occurs in adults aged 30–60 years and is a female predilection; the exact etiology remains unclear. Epithelial–mesenchymal transition (EMT) is the transdifferentiation of stationary epithelial cells primarily activated by a core set of transcription factors (EMT-TFs) involved in DNA repair and offers advantages under various stress conditions. Data have suggested that EMTs represent the basic principle of tissue heterogeneity in PAs, demonstrating the potential of adult epithelial cells to transdifferentiate into mesenchymal cells. It has also been reported that multiple TFs, such as TWIST and SLUG, are involved in EMT in PA and that SLUG could play an essential role in the transition from myoepithelial to mesenchymal cells. Given this background, this review aims to summarize and clarify the involvement of EMT in the development of PA, chondrocyte differentiation, and malignant transformation to contribute to the fundamental elucidation of the mechanisms underlying EMT.

## 1. What Is Pleomorphic Adenoma (PA)?

PA is the most common salivary gland tumor, representing up to two-thirds of all salivary gland neoplasms [1]. First termed as PA by Willis [1], it has also been referred to as branchioma, enclavoma, enchondroma, endothelioma, and mixed tumor, among others [2]. It most frequently occurs in the parotid glands (85%), followed by the minor salivary (10%) and submandibular glands (5%) [3]. The World Health Organization defines PA as a localized tumor that presents pleomorphic or mixed characteristics of epithelial origin and is interwoven with mucoid tissue, myxoid tissue, and chondroid masses.

Although PA most commonly appears in the parotid glands, it can also be located in the hard and soft palate and saliva glands of the upper lip, cheek, tongue, and floor of the mouth [4]. The morphological complexity of PA, which presents with pathognomic histopathologic features across glands and individuals, is the basis of the term. PA is a single cell that differentiates into epithelial or myoepithelial cells as opposed to multiplying carcinogenic epithelium and myoepithelium cells concurrently [5]. The recognition of PA is conceptualized by identifying three components: epithelial, myoepithelial, and mesenchymal. Histologically, PA presents as a variable epithelium pattern in a loose fibrous myxoid-, chondroid-, or mucoid-type stroma. Myoepithelial cells have a polygonal shape with pale eosinophilic cytoplasm.

Microscopic identification is needed for a definitive diagnosis of PA [6]. It is known that the incidence of PA increases from 15 to 20 years after radiation exposure. However, the exact etiology is unknown, and the cause of PA remains unclear. A few previous studies have reported an association between PA and simian virus 40 (SV40). Furthermore, cytogenetics and molecular studies have suggested an association with chromosomal aberrations involving 8q12 and 12q [7]. Moreover, the use of tobacco, exposure to chemicals, and genetic predisposition are suggested to play a role in the etiology of the disease [8]. PA typically appears as an irregular nodular lesion with a firm consistency. If the PA is superficial and does not show any fixation, areas of cystic degeneration can be palpated. PA in the minor salivary glands most frequently occurs in the palate, upper lips, and buccal mucosa [1,9] and is typically asymptomatic, painless, and does not involve the facial nerve. If no interventions are implemented in the early stages, PA can grow to massive proportions and may become malignant. In tissue sections, PA appears as an irregular ovoid mass with well-defined borders and may remain unencapsulated or be covered by an incomplete fibrous capsule. PA can have a rubbery, fleshy, or mucoid consistency interspersed with areas of hemorrhage and infarction [10].

Although computed tomography (CT) and magnetic resonance imaging (MRI) can be used to confirm the presence of the tumor, MRI is preferred. MRI allows a better delineation of the tumor margins and their location concerning surrounding tissues. However, to differentiate malignant and benign lesions, fine-needle aspiration is used. Although PA is encapsulated, it is still excised with adequate margins involving surrounding healthy tissue; this is because pseudopodic cells exhibit microscopic extensions into the surrounding tissues due to dehiscences in the false capsule. Therefore, to prevent the spillage of tumor cells, incisional biopsy is avoided [11]. Surgical excision is the most common treatment. Superficial parotidectomy with facial nerve preservation is frequently performed for the PA at the superficial lobe of the parotid gland. If the tumor involves the deep lobe, total parotidectomy is carried out. Wide local excision involving the periosteum or bone is used to treat PA in the minor salivary glands, as enucleation is associated with an increased risk of local recurrence [5]. The prognosis of PA is good, with a 95% overall cure rate. Radiotherapy is not indicated because the tumor is radio-resistant [5,12].

Clinically, PA can be diagnosed as a palatal abscess, odontogenic or nonodontogenic cyst, or soft tissue tumors such as fibroma, lipoma, neurofibroma, neurilemmoma, lymphoma, or other salivary gland tumors. A palatal abscess can be differentiated by identifying its source, such as a nonvital tooth in the immediate surroundings. Neither odontogenic nor nonodontogenic cysts show a cystic nature during an exploration into the mass [5]. Due to its varied histopathological presentation, PA can be confused with myoepithelioma, mucoepidermoid carcinoma, adenoid cystic carcinoma (ACC), basal cell adenoma, or epithelial–myoepithelial carcinoma [6,13]. Myoepitheliomas are relatively rare benign salivary gland tumors consisting of neoplastic myoepithelial cells. The ductal structure is lacking or only slightly noticeable. It tends to occur in adults, and there seem to be no gender differences. It occurs most often in the parotid glands in the large salivary glands and in the palatine glands in the minor salivary glands. Clinically, it is a slow-growing mass with bulging elastic toughness. Macroscopically, it is a well-defined solid tumor with a capsule around it. Histologically, it is classified into spindle cell type, epithelioid cell type, epithelioid cell type, clear cell type, and mixed type in which these are mixed. However, myoepithelioma does not exhibit typical features such as glanduloductal differentiation or the absence of chondromyxoid or chondroid foci [6,13]. Intermediary cells are a common feature in both mucoepidermoid carcinoma and PA. Mucoepidermoid carcinoma consists of mucus-producing cells, epithelioid cells, and intermediate cells, which are smaller in size and morphologically do not belong to either of these cells. It originates from the salivary glands, the exocrine glands in the area covered by the respiratory tract hair epithelium, and the cervix. It is one of the most common malignant salivary gland tumors. The 5-year survival rate is as good as 80%, and it has a relatively good prognosis. However, some have a poor prognosis and a low degree of differentiation. Mostly in the parotid glands, 40% in the minor salivary glands occur in the palate. It is common among women in their 30 s and 40 s. It is rare in children under 10 years of age, but it is common among malignant tumors in children. The capsule is indistinct, and infiltration into surrounding tissues is conspicuous in poorly differentiated ones. There is no pain at the beginning, but when it grows larger, it causes pain and neuropathy, and it is usually noticed within one year. It may occur in the jawbone. Histopathologically, cystically dilated ducts and irregular ducts show bright cytoplasmic cells with clear mucus production. In the surrounding area, the proliferation of flat epithelial-like cells showing the paving stone-like arrangement and intermediate type cells forming solid follicles is observed. Squamous cell-like cells show no keratinization, the stroma is fibrotic tissue, and the tumor capsule is unclear. However, in mucoepidermoid carcinoma, they produce extracellular material and cannot create myxochondroid stroma [5,6,13]. ACC is produced from exocrine glands such as lacrimal glands, salivary glands, and mammary glands, which have a structure called myoepithelial cells and have a function of actively squeezing secretions. In extremely rare cases, it may occur in organs that would not normally have myoepithelial cells, such as the uterus, and is thought to be derived from metaplastic cells or pluripotent epithelial stem cells. In salivary gland tumors, the frequency is high, the cell atypia is not high, but the infiltration tendency is strong, and the metastasis rate is high. Relapses may repeatedly occur, eventually resulting in a poor prognosis. The sieving structure is characteristic, but there are many other cases in which solidity or ductal structure is predominant. ACC has a tendency to directly invade the adjacent nerve sheaths close to the primary tumor and spread along the nerve [14]. Furthermore, it often causes neurological symptoms and may be accompanied by facial nerve paralysis. Hematogenous metastases to the lungs, bones, and skin have also been reported. It usually occurs in women around the age of 50. The recurrence rate is high, and the growth is relatively slow, but the prognosis is poor, especially in the submandibular and sublingual glands. Since there is infiltration around the nerve, it is necessary to secure a sufficient safety margin when excising. The identification of ACC is made based on its tendency for perineural invasion and infiltrative growth patterns [6,13]. Basal cell adenoma is a benign tumor that develops in the salivary gland and is a localized tumor with a clear capsule consisting of uniform proliferation of basal cell-like cells. It is relatively rare among salivary gland tumors. In particular, it is extremely rare to occur in the submandibular gland [6,13]. Basal cell adenocarcinoma is a malignant type of basal cell adenoma, but it lacks atypia and polyphasic cells and is difficult to distinguish by cytopathology alone. Histologically, the presence or absence of infiltration and proliferation to the surroundings is essential for differentiation. However, in the case of pleomorphic adenoma, there may be cases in which basal cell-like cells are the main constituents, but even in such cases, a certain number of myoepithelial cells showing other types of morphology are usually mixed, which is a clue for differentiation [6,13]. Epithelial–myoepithelial carcinoma is a tumor consisting of a follicle of myoepithelial cells in the form of clear cells and a bilayer duct, and clear cells are usually arranged in the outer layer of the duct. However, in recent years, cases with prominent basal cell-like traits and cases with peculiar images that can be called histological modifications such as apocrine-like characteristics and differentiation into sebaceous glands have been reported. In general, since it shows a monotonous image mainly composed of clear cells, it is unlikely that pleomorphic adenoma is mistaken for this tumor, but this tumor is clinically low malignant and histologically atypical is conspicuous. However, the boundaries may be relatively clear, and it is possible that this tumor may be mistaken for pleomorphic adenoma. It lacks myxomatous stroma and osteochondral, and the presence or absence of plasma cell-like cells appears to be a major indicator of differentiation [6,13,15].

In PA, malignancy occurs in three forms: mostly as carcinoma ex-pleomorphic adenoma (Ca ex PA), and rarely as carcinosarcoma and metastasizing pleomorphic adenoma (MPA) [16,17]. A systematic review of 81 cases of MPA by Knight et al. [18] found that bone, lung, and cervical lymph nodes were the most common sites for MPA, with occurrences of 36.6% (28 cases), 33.8% (26 cases), and 20.1% (17 cases), respectively; other sites included the kidneys (8.6%), cutaneous (8.6%), hepatic (4.9%), and brain (3.7%). The risk of recurrence of PA is typically associated with a poor surgical procedure, resulting in spillage of the tumor or tumor capsule. Furthermore, the recurrence of PA occurs as multiple, separate nodules. The associated surgical risks are pseudopodia, capsular penetration, and tumor rupture [19].

## 2. Epithelial–Mesenchymal Transition (EMT) in Tumor Progression

Epithelial–mesenchymal transition (EMT), first observed in early development, is a term used to describe the transdifferentiation of quiescent epithelial cells to mesenchymal and motile phenotypes [20]. EMT is known to contribute to embryonal processes such as gastrulation, heart development, and neural crest formation [21,22] and physiological processes such as wound healing [23] and tissue homeostasis [24]. In addition, pathological reactivation of EMT is known to play a fundamental role in diseases such as organ fibrosis and the progression of cancer to metastasis. Cancer is a very complex and diverse disease that varies not only between entities but also within the same entity, between different subtypes, and even within subtypes. In particular, within the same individual, tumors exhibit not only spatial heterogeneity but also temporal heterogeneity. This can be triggered by continuous mutations and clonal evolution [25]. However, the EMT process mediates the plasticity of cancer cells, allowing for continuous and reversible adaptation to constantly changing conditions. Furthermore, it is not genetically fixed because it depends on the mutations that accumulate. It is epigenetically tuned by signals from the microenvironment, making the entire program reversible (i.e., by activating mesenchymal–epithelial transformation) and highly dynamic [26].

EMT is mainly activated by a core set of transcription factors (EMT-TFs), including *SNAIL* (also *SNAI1*) and *SLUG* (also *SNAI2*), the basic helix–loop–helix factors *TWIST1* (*TWIST*) and *TWIST2*, and the zinc finger E-box binding homeobox factors *ZEB1* and *ZEB2*. All of these factors can repress epithelial genes such as the E-cadherin-encoding gene *CDH1* via binding to E-box motifs in cognate promoter regions [21], as shown regarding *SNAIL* [27,28], *TWIST* [29], *ZEB1* [30], and *ZEB2* [31]. In parallel, EMT-TF directly or indirectly activates genes associated with mesenchymal phenotypes such as *VIM* (vimentin), *FN1* (fibronectin), and *CDH2* (N-cadherin) [21,32]. However, many functions are performed by separate, unshared EMT-TFs due to differences in expression patterns and protein sizes and structures [33]. Beyond the “classical” EMT properties, EMT-TF is widely important in cancer biology, as demonstrated by its additional pleiotropic function [34]. EMT-TF helps maintain stem cell properties, enhances tumorigenicity, and links to cancer stem cells. In addition, EMT-TF provides a survival-promoting phenotype that is involved in DNA repair, antigenic escape, treatment resistance, aging, and escape from apoptosis and provides benefits under a variety of stress conditions. Altogether, the combination of classical EMT functions and the highly diverse, context-dependent, nonredundant, and nonclassical functions of EMT-TFs, which is also dynamically regulated by the tumor microenvironment, enables cancer cells to adapt permanently to changing conditions [35]. As a result, therapeutic interventions, including EMT/plasticity, are thought to help combat many aspects of tumor progression with a single blow.

We have been studying lymph node metastasis using human oral cancer cell lines. Human oral cancer cells were inoculated into the tongue of nude mice, metastasized to the submandibular lymph nodes, and oral cancer cells were isolated and cultured from the metastatic lesions to establish a highly metastatic strain. It was found that this highly metastatic cell line to the lymph node promoted EMT induction as compared with the parental line. Furthermore, it was reported that the mesenchymal marker Fibronectin induces the expression of *VEGF-C* and promotes lymphangiogenesis. This indicates that EMT of cancer cells indirectly induces lymph node metastasis of malignant tumors [36]. Further in vivo studies on the relationship between EMT of cancer cells and the tumor microenvironment are desired.

### 2.1. EMT Marker

The following is a brief description of typical EMT markers.

#### 2.1.1. Major Epithelial Markers

##### E-Cadherin

E-cadherin is an essential molecule in maintaining epithelial integrity and is involved in the regulatory mechanisms of epithelial cell proliferation, differentiation, and survival. It has also been suggested that E-cadherin may also be interested in tumorigenesis.

##### Cytokeratin

Cytokeratin is a family of intermediate filaments that provide mechanical support in epithelial cells. Cytokeratin expression is organ/tissue-specific and differentiation-dependent. Cytokeratin is used as a diagnostic tumor marker because epithelial malignancies maintain specific cytokeratin patterns associated with a cellular origin.

#### 2.1.2. Major Mesenchymal Markers

##### N-Cadherin

N-cadherin (nerve cadherin) is a 130-kDa transmembrane glycoprotein, also known as CDH2 (cadherin 2), and is one of the classic members of the cadherin superfamily. The expression of N-cadherin has been reported in various cells, including neurons, endothelial cells, and cardiomyocytes.

##### Vimentin

Vimentin is an intermediate filament unique to mesenchymal cells. Vimentin is a major cytoskeletal protein distributed in various cells such as fibroblasts, vascular endothelial cells, smooth muscle cells, collateral muscle cells, bone/cartilage cells, and nerve sheath cells that make up the connective tissue.

##### Fibronectin

Fibronectin is a glycoprotein that forms an extracellular matrix, and a polypeptide with a molecular weight of about 250 kDa forms a dimer. It mainly promotes the adhesion of fibroblasts, hepatocytes, nerve cells, etc. Integrin, a specific receptor on the surface of cell membranes, is involved in cell adhesion, cell migration, phagocytosis, etc. It works in the field of tissue damage.

## 3. EMT in PAs

Frequently, PA involves areas in which myoepithelial cells lose adhesion and disperse in copious chondroid/myxoid stroma; this has been recognized as EMT [37,38]. As a feature of PA, Masson favored mesenchymatous transformation, which may have been influenced by his investigations on Wilms tumor, where similar transformations occur [38,39,40]. In this process, which was subsequently described as mesenchymalization or stromalization, attributable to the activation of dormant mesenchymal genes in tumor epithelial cells [41], formerly polarized tumor epithelial cells lose cell adhesion molecules (E-cadherin) and secrete matrix [42,43] before separating and dispersing in the copious myxoid stroma, where they have been observed to simulate primitive mesenchyme or “swarming bees” and to express α5-integrin, Fibroblastic and chondrocyte collagens (types I–III) [44,45].

Mesenchymalization/romanization, as described above, falls within the range of EMT [46]. Immunohistochemistry has revealed that in PA, much of the tumor parenchyma shows transitional, epithelial, and mesenchymal phenotypes [37]. The aggrecan (chondroitin sulfate proteoglycan 1) and *CK14* mRNAs are localized in luminal and non-luminal cells of the epithelial and mesenchymal phenotypes, respectively, by in situ hybridization [37,47]. The variable immunohistochemical localization of transforming growth factor (TGF)-β isoforms in luminal and non-luminal tumor cells is further supported by EMT in PA [48] because TGF-β affects EMT [49]. EMT can also account for the hyalinization/collagenous structures, elastosis and cartilaginous, osseous, myoid (smooth muscular), and adipocytic [50,51] phenotypes of PA, which in turn, explains its complex microstructure. Therefore, assessing the expression of Snail1 (a protein that influences EMT through the transcriptional repression of E-cadherin) in PA is of interest. As a feature of PA, EMT is most undoubtedly appealing, but the argument that PA does not originate in the exocrine pancreas, where myoepithelial cells are absent, anchors and reinforces the notion of neoplastic or modified myoepithelial [52]. Langman et al. [53] reported that Wilms tumor 1 protein (WT1) is co-expressed by calponin (+) and p63 (+) in non-luminal cells in PA, suggesting the usefulness of WT1 as a myoepithelial marker. Interestingly, they did not report any WT1 immunoreactivity in normal salivary myoepithelial cells [53]. A more recent study confirmed the absence of WT1 immunoreactivity and suggested that WT1 (+) cells undergo EMT in PA [54], which would be consistent with the role played by the WT1 gene in affecting epithelial or mesenchymal status [55]. Caution is needed before interpreting the immunoreactivities of purported pathological analogs as markers when particular macromolecules are not expressed in normal cells. The trend of discovering novel myoepithelial features continues, with podoplanin being a recent example [56].

EMT and neoplastic or modified myoepithelial may not be mutually exclusive. In a tumor cell undergoing myoid EMT, a loss of intercellular cohesion and the cytoplasmic accumulation of myofibrils would be expected, thereby qualifying the entity as a neoplastic myoepithelial or modified myoepithelial cell [54]. However, EMT allows a broader perspective and explains non-myoid cell phenotypes separating from the intervening phase of modified myoepithelium [38]. Although outside the scope of the present article, “myoepitheliomas” are considered members of the PA family [57], as it is likely that they are PAs that feature widespread myoid EMT, eventually resulting in the “depletion” of luminal structures [38]. PAs of the parotid gland were analyzed by Aigner et al. [37] as a model that shows morphological features of epithelial and mesenchymal tissue types. They demonstrated areas with unequivocal epithelial and mesenchymal differentiation by using matrix gene expression profiles as an additional criterion to identify cellular phenotypes. Many regions showed a transitional phenotype, with cells demonstrating epithelial and mesenchymal features. These data suggested that EMTs represent the basic principle of tissue heterogeneity in PAs and concluded that PAs illustrate the potential of adult (neoplastic) epithelial cells to transdifferentiate into mesenchymal cells in vivo.

## 4. EMT-Activating Transcription Factors (EMT-TFs) in PAs

### 4.1. TWIST and SLUG Are Expressed as EMT-TFs in PAs

Histologic diversity due to myoepithelial cells with morphologic plasticity, which can be attributed to EMT, is the hallmark of PAs. The occurrence of EMT within PA has been demonstrated by immunohistochemical and ultrastructural analyses [48,58,59], but no specific TFs have been identified. To our knowledge, only four studies have investigated the expression of EMT-TFs in PAs through the examination of *TWIST* or *SLUG* expression by immunohistochemical or reverse transcription-quantitative polymerase chain reaction (RT–qPCR) analysis [60,61,62,63].

Pardis et al. [60] reported observing TWIST expression in 12 cases of PA, predominantly with a cytoplasmic pattern and moderate intensity, evaluated immunohistochemically. Furthermore, Yuen et al. [64] reported that the cytoplasmic expression of TWIST was associated with neoplastic transformation in prostate tissues. In Pardis et al.’s study, the overexpression of *TWIST* also seemed to be related to neoplasm formation in the salivary glands. In agreement with Pardis et al., Shen et al. [62] identified the overexpression of *TWIST* in 30% of PAs, which had only previously been demonstrated in some benign and malignant tumors. TWIST had been observed in the parenchymal cells of benign tumors of the prostate, parathyroid, and lung and precancerous lesions of the oral cavity [60,64,65,66,67].

### 4.2. TWIST1 Inhibits Chondrocyte Differentiation in PAs

We have previously provided conclusive but indirect evidence that epithelial cells differentiate into chondrocytes during salivary gland PA tumorigenesis [61]. Our study found that epithelial cells express Sox9 and Sox6 and produce aggrecan and type II collagen, which are substances in the extracellular matrix peculiar to cartilage. Sox9, Sox6, and Sox5 make up a trio of TFs essential for chondrocyte differentiation. Sox9 functions in concert with Sox6 or Sox5 as a master regulator of chondrocyte differentiation [37]. The TF of this trio induces chondrocyte differentiation in both chondrogenic and non-chondrogenic mesenchymal cells already involved in other lineages, such as epithelial cells from the cervix, liver, and kidneys [68]. Both human and mouse salivary gland cells have been shown to express Sox9 [59], as well as ductal and acinar cells. These cells are similar to PA epithelial cells because they express Sox6. However, it differs from PA because it does not produce genes that are not transcribed in the salivary glands, such as aggrecan and type II collagen. The most crucial step in controlling gene expression is RNA transcription, which depends on the balance of positive and negative TFs. Negative TF suppresses positive TF when cartilage-specific genes such as *δEF1*, *AP-2α*, *SNAIL*, *SLUG*, *Twist1*, and *C/EBPβ* [48,69,70,71,72] are not expressed. Among these genes, *δEF1*, *AP-2α*, and *Twist1* are expressed by chondrocyte progenitor cells and can remain undifferentiated [58,69,71,72]. Therefore, their expression in salivary glands was compared to that in PA. The mRNAs of *δEF1* and *AP-2α* were detected in both salivary glands and PA. *Twist1* mRNA was found in the salivary glands but not in three of the four PAs. Based on the results of RT-qPCR analysis, significantly less *Twist1* mRNA was found in the remaining tumors than in the salivary glands. Based on immunohistochemistry results, the Twist1 and Sox9 proteins were localized to the same salivary gland cells. These results suggest that Twist1 expression can suppress the potential transactivation of the Sox protein in salivary gland cells. In addition, the depletion of Twist1 that occurs during the neoplastic transformation of salivary gland cells allows the Sox protein to transcribe the aggrecan and type II collagen genes. *Twist1*, a member of the TF’s basic helix-loop-helix family, is essential for the development of tissues of mesoderm origin, emphasizing its function in EMT and metastasis [73]. *Twist1* has also been identified as a downstream mediator of standard Wnt signaling, known to suppress cartilage cell differentiation in cartilage [72]. Furthermore, ectopic expression of *Twist1* suppressed the expression of chondrocyte marker genes such as type II collagen and aggrecan in mouse chondrocyte progenitor cells. In contrast, *Twist1* depletion enhanced the expression of these genes [72]. *Twist1* can directly or indirectly regulate the expression of the target gene. During chondrocyte differentiation, it acts indirectly by binding to Sox9 and blocking its transactivation potential [58]. It also binds to MyoD and Runx2, the master transcriptional regulators of myogenesis and bone formation, respectively, and inhibits the differentiation of mesenchymal precursor cells into these lines [73]. Therefore, we hypothesized that Twist1 interferes with the differentiation of salivary gland cells into chondrocytes. To test this hypothesis, we conducted in vitro gain-of-function and loss-of-function experiments with Twist1 to examine the expression of aggrecan and type II collagen in human submandibular gland (HSG) cells [74]. Although HSG cells are neoplastic when inoculated into immunodeficient mice, they have been found to retain many of the characteristics of PA-generating salivary duct cells [74]. We also found that HSG cells contained the elements of salivary duct cells. They expressed Sox9, Sox6, and Twist1, but not aggrecan or type II collagen. The knockdown of Twist1 by small interfering RNA resulted in upregulation of both aggrecan and type II collagen gene expression. In contrast, overexpression of Twist1 led to the downregulation of these two genes. These results supported our hypothesis [61].

### 4.3. SLUG Is an Important EMT-TF for EMT Induction of Myoepithelial Cells in PA

The localization of *TWIST1* expression has also varied (i.e., nuclear vs. cytoplasmic) between studies. To improve the sensitivity and specificity for detecting the expression of EMT-TFs, including *SNAIL*, *SLUG*, *ZEB1*, and *TWIST1*, Kim et al. used RNA in situ hybridization (ISH) as opposed to immunohistochemistry in a series of PAs [63]. They also investigated the association between *PLAG1* and *SLUG* expression and the functional roles of *SLUG* in EMT using primary cultured PA cells and the expression of four significant EMT-TFs. As a result, they found that *SLUG* was upregulated in PAs and restricted its expression to neoplastic myoepithelial and stromal cells. Furthermore, using primary cultured PA cells reported that *SLUG* was involved in tumor growth and the regulation of EMT marker expression, revealing that *SLUG* was a significant TF in EMT in PAs.

In contrast to negligible *SNAIL*, *ZEB1*, and *TWIST1* expression, they found low but significant levels of *SLUG* expression in normal salivary glands according to RT–qPCR analysis. Using a mixed approach by combining RNA ISH for *SLUG* and multiplex immunohistochemistry for *CK19* and *P63* allowed them to observe the co-localization of *SLUG* and *P63*, confirming that myoepithelial cells in the salivary glands express *SLUG* commonly. Their findings suggested that *SLUG* is a critical TF that confers mesenchymal features such as contractile function and elongated morphology to myoepithelial cells, which are present in Bartholin’s glands and the mammary, sweat, lacrimal, and mucous glands of the aerodigestive tract [68]. Furthermore, these findings concluded that the expression of *SLUG* in the myoepithelium of other organs should be examined. Indeed, Guo et al. [75] found that the Slug protein is specifically expressed in basal cell nuclei in murine mammary epithelium and that Slug and Sox9 act cooperatively to determine the mammary stem cell state. Kim et al. [63] reported that myoepithelial cells start to separate and disperse into the myxoid stroma at the periphery of tumor glands in PAs, in contrast to cohesive luminal structures. They did not observe SLUG expression in luminal epithelial cells. Only SLUG-positive myoepithelial cells were observed to exhibit changes in EMT markers, i.e., the downregulation of E-cadherin and the upregulation of N-cadherin and vimentin. In addition, no histological evidence has been found to suggest a direct transition from luminal epithelial to stromal cells in PAs. Therefore, EMT in PAs can technically be described as a transition from myoepithelial to mesenchymal cells. However, luminal epithelial cells may have the potential to transform into mesenchymal cells. It remains unclear whether a change between luminal epithelial and myoepithelial cells can occur under certain circumstances in PAs.

These findings suggest that multiple TFs, including *TWIST* and *SLUG*, are involved in EMT in PA. That *SLUG* may play an essential role in the transition from myoepithelial to mesenchymal cells. Furthermore, there seems to be a complicated mechanism in which *TWIST* expression is reduced to induce the differentiation into chondrocytes (Figure 1).

## 5. Carcinoma Ex-Pleomorphic Adenoma (Ca ex PA) and EMT

A characteristic of EMT, mediated by the signaling molecule TGF-β1 [76], is the loss of E-cadherin expression. In general, changes in gene expression are achieved primarily by genetic and epigenetic methods. Genetic changes broadly alter the structure or number of specific genes, whereas epigenetic changes occur at the transcriptional level [77]. The standard epigenetic method for modifying gene expression is methylation of CpG islands in the promoter region. CpG methylation is primarily associated with various types of cancer, including squamous cell carcinoma of the esophagus (SCC) [78,79], oral SCC [80], salivary Ca ex PA [81], and ACC [82]. It regulates tumor progression through the inactivation of tumor suppressor genes such as *p16*, *MGMT*, *DAPK*, and *RASSF1A*. Changes in the methylation status of the *CDH1* promoter have been reported to be essential contributors to E-cadherin silence in many tumors [77,83,84], and *CDH1* silence is the stage and attack of advanced tumors. It is directly related to the typical expression type [83].

Xia et al. [85] first investigated the methylation status of the *CDH1* promoter in salivary Ca ex PA. They found a link between the methylation of the *CDH1* promoter and the expression of E-cadherin in 35.14% (13/37) of cases of Ca ex PA in terms of the absence of expression of E-cadherin. Similar to the findings reported by Zhang et al. [86], a negative E-cadherin detection rate of 38.33% was observed in 60 cases of CAC. Nevertheless, they also found negative E-cadherin expression in 68.42% (26/38) of eyelid SCCs and 87.26% (18/23) of oral SCC cases. In another study [83], a 42.33% (58/137) rate of decreased E-cadherin expression was found in breast cancer. Therefore, decreased E-cadherin expression appears to occur at varying frequencies in different tumor types and relatively infrequently, especially in salivary gland tumors. In the same study, methylation of the *CDH1* promoter was also detected using bisulfite sequencing PCR. It can detect methylation at each CpG site individually and is considered the “gold standard” for determining DNA methylation. They found a *CDH1* methylation rate of 67.57% (25/37) in Ca ex PA, similar to many other tumors such as primary lung cancer (88%) [87], breast carcinoma (65–95%) [83,88,89,90], and colorectal carcinoma (52%) [91]. They also reported that DNA methylation occurred preferentially in the first four compared to other CpG islands. Furthermore, they investigated the link between the methylation state of *CDH1* and the expression of E-cadherin in patients with Ca ex PA. Methylation of *CDH1* was significantly associated with decreased expression of E-cadherin in clinical samples (*p* < 0.001). In addition, they evaluated *CDH1* mRNA and protein levels corresponding to *CDH1* methylation status in two Ca ex PA cell lines, SM-AP1 and SM-AP4. According to the above results, they found that cells with higher levels of *CDH1* methylation had a weaker E-cadherin expression. In addition, TGF-β1 treatment of SM-AP1 cells led to the downregulation of E-cadherin and the upregulation of vimentin in vitro experiments, suggesting that EMT might play a character in the repression of E-cadherin in salivary Ca ex PA.

In spite of these results, no association was found between the methylation of the *CDH1* promoter and the downward regulation of e-cadherin expression levels. Many studies have shown that *CDH1* expression may be suppressed by mechanisms other than the methylation of the promoter. These include chromatin structural changes, loss of heterozygosity at 16q22.1, inactivated genetic mutations, specific transcription factors, and translational and posttranslational regulation [92,93,94]. Xia et al. concluded that E-cadherin expression levels are regulated primarily by DNA methylation of Ca ex PA, both in vivo and in vitro. They also stated that tumors with methylation of the *CDH1* promoter showed the following histopathological tendencies: Lumen differentiation (*p* = 0.004), high tumor grade (*p* = 0.005), high T stage (*p* = 0.024), and high TNM stage (*p* = 0.038). In other words, it was suggested that the malignancy of Ca ex PA increases due to the decreased expression of E-cadherin. Still, they are not the only other regulators that affect *CDH1* in Ca ex PA. It suggests that the mechanism needs to be investigated in future studies.

## 6. Conclusions

Pleomorphic adenoma is the most frequently occurring tumor of the salivary gland. This tumor is benign but can have a relapse after an incomplete excision and can turn into a malignant tumor.

Pleomorphic adenoma is an encapsulated tumor, but the capsule may be incomplete or show infiltration by a tumor. CT and MRI are used for diagnostic imaging, and puncture aspiration biopsy is preferred. PA is confirmed by histopathological examination. It is characterized by a mixed appearance of epithelial, myoepithelial, and mesenchymal components. The mesenchymal component may include mucoid tissue, myxoid tissue, and infrequently chondroid tissue. From several ultrastructural and immunohistochemical studies, it was suggested that the mesenchymal component is epithelial in origin.

Although EMT is highly likely to be involved in developing PA, chondrocyte differentiation, and malignant transformation, many points remain unclear, and further research seems to be required. A more detailed study of EMT in PA would be expected to contribute to the fundamental elucidation of the mechanisms underlying EMT.

## Figures and Tables

**Figure 1 jcm-11-04210-f001:**
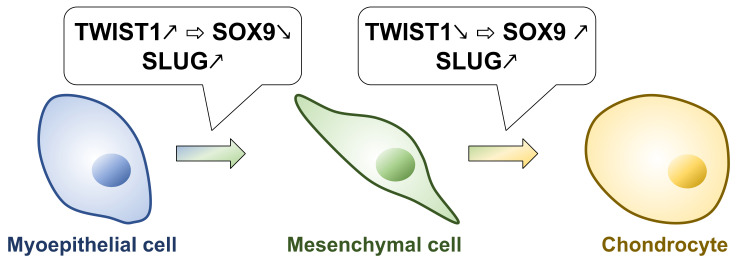
EMT induction and chondrocyte differentiation by changes of TFs expression in PAs.

## Data Availability

No new data were created or analyzed in this study. Data sharing is not applicable to this article.

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
