# Peer review of "Pleomorphic Adenoma of the Salivary Glands and Epithelial–Mesenchymal Transition"

_jcm, 2022, doi:10.3390/jcm11144210_

Round 1

Reviewer 1 Report

Transparent manuscript scheme, differential diagnosis discussed in detail, EMT involvement in PA development, chondrocyte differentiation and malignant transformation comprehensively explained.

Noteworthy too limited number of citations above 2015.  

Author Response

The authors are grateful to Reviewer #1 for the careful reading of our manuscript and the constructive and encouraging comments. We have revised our manuscript in response to Reviewer #1’s comments. Stated below are our replies to each of the comments made by Reviewer #1. We hope that these replies will meet the requirements of Reviewer #1 and that the revised manuscript is acceptable for publication in the Journal.

Transparent manuscript scheme, differential diagnosis discussed in detail, EMT involvement in PA development, chondrocyte differentiation and malignant transformation comprehensively explained.

Noteworthy too limited number of citations above 2015. 

  • Thank you for your kind review. As you pointed out, research on EMT in cartilage differentiation of PA has not progressed much since 2016. We are currently continuing research from clinical samples.

Reviewer 2 Report

This article provides an interesting insight into EMT in the context of pleomorphic adenoma, treating the argument in a specialistic approach. Personally, I appreciate such a narrative review design in those cases where an extended and complex topic has to be addressed.

My only suggestion is about the introduction paragraph.

I suggest to better summarize the selected informations. Taking in mind the topic of the review, it might be misleading to discuss all the aspects related to the differential diagnosis, clinical presentation, diagnostic workflow and so on. Some of these information could be removed in order to provide a quicker introduction leading to the main core of the paper.

Author Response

The authors are grateful to Reviewer #2 for the careful reading of our manuscript and the constructive and encouraging comments. We have revised our manuscript in response to Reviewer #2’s comments. Stated below are our replies to each of the comments made by Reviewer #2. We hope that these replies will meet the requirements of Reviewer #2 and that the revised manuscript is acceptable for publication in the Journal.

I suggest to better summarize the selected informations. Taking in mind the topic of the review, it might be misleading to discuss all the aspects related to the differential diagnosis, clinical presentation, diagnostic workflow and so on. Some of these information could be removed in order to provide a quicker introduction leading to the main core of the paper.

  • We considered that some clinical information was needed in the PA review. However, it seems that there was too much clinical information, so I removed some sentences.

Reviewer 3 Report

Thank you for this nice and well-written review on pleomorphic adenoma  in salivary glands with a specific focus on EMT in relation to this disease.

Please allow me following comments: 

There is some redundancy in the text about 1. What is pleomorphic adenoma…especially about facial nerve affection

“ Neither odontogenic nor nonodontogenic cysts show a cystic nature” Parts of this chapter would better fit into the differential diagnosis section

about adenoid cystic carcinoma: typically one talks about a neurophilic tumor: often nerve involvement is seen and is a typical mechanism of tumor spread

Why do you concentrate on these 4 differential diagnoses: Myoepithelioma/Mucoepidermoid/ACC/ BCA(C)?

Concerning the EMT chapters:

figure 1 shows a hypothesis of how EMT induction and chondrocyte differentiation could happen. As stated in the text other TFs are involved and maybe this should be introduced in the figure to give a better overall picture.

The conclusion is rather short. Maybe a bit more could be included.

Author Response

The authors are grateful to Reviewer #3 for the careful reading of our manuscript and the constructive and encouraging comments. We have revised our manuscript in response to Reviewer #3’s comments. Stated below are our replies to each of the comments made by Reviewer #3. We hope that these replies will meet the requirements of Reviewer #3 and that the revised manuscript is acceptable for publication in the Journal.

There is some redundancy in the text about 1. What is pleomorphic adenoma…especially about facial nerve affection.

  • We have revised the manuscript in the introduction section to reduce the description of the facial nerve.

“ Neither odontogenic nor nonodontogenic cysts show a cystic nature” Parts of this chapter would better fit into the differential diagnosis section

  • We have revised the differential diagnosis section and changed the manuscript.

About adenoid cystic carcinoma: typically one talks about a neurophilic tumor: often nerve involvement is seen and is a typical mechanism of tumor spread

  • We have revised the manuscript on adenoid cystic cancer.

Why do you concentrate on these 4 differential diagnoses: Myoepithelioma/Mucoepidermoid/ACC/ BCA(C)?

  • We have listed the typical differential diagnosis of PA in the manuscript. In addition to these, we added epithelial-myoepithelial carcinoma and revised the manuscript of differential diagnosis.

Concerning the EMT chapters:

figure 1 shows a hypothesis of how EMT induction and chondrocyte differentiation could happen.

As stated in the text other TFs are involved and maybe this should be introduced in the figure to give a better overall picture.

  • The involvement of other TFs is not so conclusive, but SOX9 is included in the figure as it has been reported to be associated with TWIST.

The conclusion is rather short. Maybe a bit more could be included.

  • We have revised the manuscript in the conclusion section.

This manuscript is a resubmission of an earlier submission. The following is a list of the peer review reports and author responses from that submission.